# Impact of preterm birth on brain development and long-term outcome: protocol for a cohort study in Scotland

James P Boardman ![ORCID],[1,2] Jill Hall,[1] Michael J Thrippleton,[3] Rebecca M Reynolds,[4] Debby Bogaert,[5] Donald J Davidson,[5] Jurgen Schwarze,[5] Amanda J Drake,[4] Siddharthan Chandran,[2] Mark E Bastin,[2] Sue Fletcher-Watson[2]

[1]MRC Centre for Reproductive Health, The University of Edinburgh, Edinburgh, UK
[2]Centre for Clinical Brain Sciences, The University of Edinburgh, Edinburgh, UK
[3]Edinburgh Imaging, The University of Edinburgh, Edinburgh, UK
[4]Centre for Cardiovascular Science, The University of Edinburgh, Edinburgh, UK
[5]Centre for Inflammation Research, The University of Edinburgh, Edinburgh, UK

**Correspondence to**
Professor James P Boardman;
James.Boardman@ed.ac.uk

## ABSTRACT

**Introduction** Preterm birth is closely associated with altered brain development and is a leading cause of neurodevelopmental, cognitive and behavioural impairments across the life course. We aimed to investigate neuroanatomic variation and adverse outcomes associated with preterm birth by studying a cohort of preterm infants and controls born at term using brain MRI linked to biosamples and clinical, environmental and neuropsychological data.

**Methods and analysis** Theirworld Edinburgh Birth Cohort is a prospective longitudinal cohort study at the University of Edinburgh. We plan to recruit 300 infants born at <33 weeks of gestational age (GA) and 100 healthy control infants born after 37 weeks of GA. Multiple domains are assessed: maternal and infant clinical and demographic information; placental histology; immunoregulatory and trophic proteins in umbilical cord and neonatal blood; brain macrostructure and microstructure from structural and diffusion MRI (dMRI); DNA methylation; hypothalamic–pituitary–adrenal axis activity; social cognition, attention and processing speed from eye tracking during infancy and childhood; neurodevelopment; gut and respiratory microbiota; susceptibility to viral infections; and participant experience. Main analyses include creation of novel methods for extracting information from neonatal structural and dMRI, regression analyses of predictors of brain maldevelopment and neurocognitive outcome associated with preterm birth, and determination of the quantitative predictive performance of MRI and other early life factors for childhood outcome.

**Ethics and dissemination** Ethical approval has been obtained from the National Research Ethics Service (NRES), South East Scotland Research Ethics Committee (NRES numbers 11/55/0061 and 13/SS/0143 (phase I) and 16/SS/0154 (phase II)), and NHS Lothian Research and Development (2016/0255). Results are disseminated through open access journals, scientific meetings, social media, newsletters anda study website (www.tebc.ed.ac.uk), and we engage with the University of Edinburgh public relations and media office to ensure maximum publicity and benefit.

## Strengths and limitations of this study

► Three hundred preterm infants and a comparator group of 100 term controls are studied longitudinally from before birth to school age.
► Phenotypical information includes data from brain MRI, biosamples, participant report, direct observation and clinical data from maternal and infant medical records.
► We collected data about a range of theoretically informed variables to understand the impact of preterm birth on everyday lives of families.
► A data access and collaboration policy sets out the terms and conditions on which deidentified data are available to the research community.
► Participants are recruited from a single centre.

equates to 14.84 million births per annum.[1] In resource-rich settings, advances in perinatal care and service delivery have led to improved survival over the past two decades: around 30% of infants born at 22 weeks who are offered stabilisation at birth will survive, and this number increases to around 80% for births at 26 weeks.[2–5] However, early exposure to extrauterine life can impact brain development and is closely associated with long-term intellectual disability, cerebral palsy, autism spectrum disorder, attention deficit hyperactivity disorder, psychiatric disease, and problems with language, behaviour and socioemotional function (for review, see Johnson and Marlow[6]). There are no treatments that reduce risk of impairment, which extends across the life course and carries considerable personal cost to affected individuals, and high health and education costs to society.[7]

Little is known about the ontogenesis of neurocognitive and psychiatric problems associated with preterm birth, or the biological, environmental and social risk factors associated with susceptibility and resilience. Much information about the cerebral effects

## INTRODUCTION

Preterm delivery is estimated to affect 10.6% of all live births around the world, which

of preterm birth comes from historic cohorts that do not reflect modern perinatal care practices; studies have been cross-sectional with outcomes assessed in very early childhood before important cognitive and social functions emerge; conventional diagnostic tools for assessing neurodevelopment are imprecise; and cohorts linked to imaging and biological metadata are few, so mechanisms are poorly understood. There is an unmet need to study a contemporary cohort of preterm infants that is comprehensively characterised from genes to anatomy to function, integrated with information about the social graph.

Our aims were, first, to build a longitudinal cohort of preterm infants and term controls that is phenotyped with brain imaging and biological information to investigate causal pathways to, and consequences of, atypical brain development and injury; second, to develop novel computational algorithms for mapping brain growth and connectivity in early life; third, to identify new and multifactorial methods for early detection of children at risk of long-term impairment; and fourth, to identify early life biological and environmental risk and resilience factors that affect the developing brain and so pave the way for new therapeutic strategies.

## METHODS AND ANALYSIS
### Study design
This is a single-centre prospective longitudinal cohort study.

### Study setting
The Theirworld Edinburgh Birth Cohort (TEBC) study is conducted at the University of Edinburgh and the Simpson Centre for Reproductive Health (SCRH), which is located at the Royal Infirmary of Edinburgh, NHS Lothian, UK. The SCRH provides maternity and newborn services for residents of the City of Edinburgh and the Lothians. It receives 7000 deliveries per annum and is the regional centre for all neonatal intensive care in South East Scotland. Approximately 100 infants with birthweight of <1500 g receive intensive care at SCRH per annum.

Participant recruitment, initial assessment and data collection points 1–3 (table 1) take place in the SCRH or the Edinburgh Imaging Facility, Royal Infirmary of Edinburgh. Follow-up assessments take place in a dedicated child development laboratory at the University of Edinburgh, through online and in-person completion of questionnaires, and in neonatal outpatient clinics at the SCRH (time points 4–7, table 1). Recruitment began in November 2016 and is planned to be completed in 2021.

### Study participants
#### Inclusion criteria
Cases: 300 preterm infants born at <33 weeks of gestational age (GA)*
 Controls: 100 term infants born at >37 weeks of GA*
 *GA is estimated based on first trimester ultrasound.

Cases are included if a mother booked her pregnancy and delivered at SCRH (the study centre), or if a mother booked her pregnancy at a hospital outside the study centre but was transferred to it with her baby in utero due to planned or expected birth <33 weeks. Preterm infants who are transferred to SCRH ex utero for intensive care are not included.

#### Exclusion criteria
1. Infants with congenital anomalies: structural or functional anomalies (eg, metabolic disorders) that occur during intrauterine life and can be identified prenatally, at birth or later in life (WHO definition).
2. Infants with a contraindication to MRI at 3T.

### Sample selection and recruitment
#### Sample size
A key aim of the study was to investigate causes and consequences of preterm brain injury/atypical development by analysing data about brain macrostructure and microstructure from structural and quantitative MRI with biological, environmental and neuropsychological outcome data. In the absence of established methodology for power calculations using quantitative MRI techniques, the sample size is based on: exemplars of indicative sensitivity and power from computational modelling and previous data; and realistic assessment of recruitment, successful image acquisition of 85%, and follow-up. Studies indicate it is possible to detect groupwise differences in brain anatomy associated with specific exposures by applying computational techniques to MRI data from relatively small group sizes in univariate models; for example, tract-based spatial statistics and network-based statistics are sensitive to generalised changes in microstructure and connectivity with 20–60 infants per group,[8–14] and morphometric methods detect anatomical variation with similar group sizes, depending on the image feature of interest.[15 16] However, a key strength of the study is that larger samples (n=300–400) are required to construct multivariate models (needed to investigate multiple exposures that influence brain development), to combine information from different MRI modalities using data-driven methods, to investigate associations between image phenotypes and behavioural outcomes which often require larger study populations,[17 18] and to develop analytical methods that support causal inference. Another aim was the development of novel computational methods for mapping growth and connectivity in development. While certain technical developments such as image segmentation and methods for studying crossing fibres are achievable with sample sizes of <100,[19–22] larger sample sizes are needed to address other challenges. For example, larger atlases of the developing brain than are currently available are required to understand population diversity, and machine learning methods are being used to develop image biomarkers and to improve the interoperability of multisite acquisitions, which will enable researchers to increase study power, carry out essential

**Table 1** Schedule of assessments, data collection methods, sample type/domain, and the test or task

| Time point | Age | Data collection method | Sample type/domain of measurement | Test/task |
|---|---|---|---|---|
| 1 | Antenatal | Records and interview | Socioeconomic status | Maternal and paternal education, Scottish Index of Multiple Deprivation derived from home postcode |
| | | | Medical/demographic | Family and medical history and exposures |
| 2 | Birth | Records, questionnaire and tissue | Medical | History and exposures |
| | | | | Anthropometry |
| | | | Placenta | Structured histopathology rating and storage |
| | | | Cord blood | Panel of immunoregulatory and trophic proteins |
| | | | | Gene expression array* |
| 3 | Neonatal | Tissue: blood | Blood spot | Panel of immunoregulatory and trophic proteins |
| | | | | Gene expression array* |
| | | Tissue: saliva | Epigenetics | DNA methylation |
| | | Tissue: nasal swab | Nasal lining fluid | Antimicrobial peptides including cathelicidin levels* |
| | | | DNA/RNA | Respiratory microbiota* |
| | | Stool | DNA/RNA | Gut microbiota* |
| | | Direct observation | Medical | Anthropometry |
| | | | Retinopathy of prematurity assessment | Grade retinopathy |
| | | | Parent IQ | National Adult Reading Test |
| | | MRI | Brain structure and connectivity | Structural and diffusion 3T MRI |
| | | Questionnaire | Medical/demographic | Breast feeding and updated perinatal medical history |
| | | | | Edinburgh Postnatal Depression Scale |
| | | | | Parenting Daily Hassles |
| | | | | WHO—Quality Of Life |
| | | | | Adult Temperament Questionnaire |
| 4 | 4.5 months | Questionnaire, by post or online or phone interview | Demographics | Updated socioeconomic status, maternal education, breastfeeding/nutrition activities |
| | | | Infant temperament | Infant Behaviour Questionnaire, Revised, short form |
| | | | Parent well-being | Edinburgh Postnatal Depression Scale |
| | | | | WHO—Quality Of Life |
| | | Tissue: nasal swab | DNA/RNA | Respiratory microbiota* |

**Table 1** Continued

| Time point | Age | Data collection method | Sample type/domain of measurement | Test/task |
|---|---|---|---|---|
| 5 | 9 months | Tissue: saliva | Epigenetics | DNA methylation |
| | | | HPAA | Cortisol: waking, 30 min after waking, before bed |
| | | | | Prestill-face and poststill-face procedure |
| | | Tissue: nasal swab | Nasal lining fluid | Antimicrobial peptides including cathelicidin levels* |
| | | | DNA/RNA | Respiratory microbiota* |
| | | Eye tracking | Social development | Free scanning: neutral faces |
| | | | | Free scanning: pop-out task, looking to faces and distractors |
| | | | | Free scanning: social preferential looking to social and non-social images |
| | | | | Free scanning: dancing ladies social and non-social videos |
| | | | Attention | Switching and disengagement: gap-overlap task, fixation to central and peripheral cues |
| | | | | Sustained attention: follow the bird task, following moving target |
| | | | Processing speed | Free scanning: odd-one-out visual search task (simple letters version) |
| | | | | Free-scanning: word–picture matching task |
| | | Direct observation | Visual acuity | Keeler card assessment |
| | | | Social development | Still-face procedure (sub-set with computational motor assessment) |
| | | | | Parent–child play, for later behavioural coding: (subset with computational motor assessment) |
| | | Questionnaire | Infant temperament | Infant Behaviour Questionnaire, Revised, short form |
| | | | | Sleep and Settle Questionnaire |
| | | | Language | MacArthur Communicative Development Inventory (words and gestures) |
| | | | Parent well-being | WHO—Quality Of Life |
| | | | Feedback | Feedback form, monitoring satisfaction with research project |
| | | Direct observation | Anthropometry | Growth |
| | | Parent interview | Demographics | Family circumstances update form including breastfeeding, socioeconomic status (home postcode) |
| | | | Developmental level | Vineland Adaptive Behaviour Scales: comprehensive interview form |

Continued

**Table 1** Continued

| Time point | Age | Data collection method | Sample type/domain of measurement | Test/task |
|---|---|---|---|---|
| 6 | 2 years | Direct observation | Ophthalmology | Refraction |
| | | | Anthropometry | Growth |
| | | Tissue: nasal swab | Nasal lining fluid | Antimicrobial peptides including cathelicidin levels* |
| | | | DNA/RNA | Respiratory microbiota* |
| | | Eye tracking | Social development | Free scanning: neutral faces |
| | | | | Free scanning: pop-out task, looking to faces and distractors |
| | | | | Free scanning: social preferential looking to social and non-social images |
| | | | | Free scanning: dancing ladies social and non-social videos |
| | | | Attention | Switching and disengagement: gap-overlap task, fixation to central and peripheral cues |
| | | | | Sustained attention: follow the bird task, following moving target |
| | | | | Free scanning: odd-one-out visual search task |
| | | | Processing speed | Free-scanning: word–picture matching task |
| | | Direct observation | Social development | Parent–child play, for later behavioural coding |
| | | | Executive function | Following Instructions task |
| | | | Bayley-III | General developmental level* |
| | | Questionnaire | Temperament | Early Childhood Behaviour Questionnaire, Revised, short form |
| | | | | Child Sleep Habits Questionnaire |
| | | | Language | MacArthur Communicative Development Inventory (words and sentences) |
| | | | Social development | Quantitative Checklist for Autism in Toddlers |
| | | | Executive function | BRIEF-P |
| | | | | Early Executive Function Questionnaire |
| | | | Developmental level | Vineland Adaptive Behaviour Scales: comprehensive parent rating form |
| | | | Parent well-being | WHO—Quality Of Life |
| | | | Feedback | Feedback form, monitoring satisfaction with research project |
| | | Parent interview | Demographics | Family circumstances update form including breastfeeding, socioeconomic status (home postcode) |

Continued

**Table 1** Continued

| Time point | Age | Data collection method | Sample type/domain of measurement | Test/task |
|---|---|---|---|---|
| 7 | 5 years | Tissue: saliva | Epigenetics | DNA methylation |
| | | | HPAA | Cortisol |
| | | Tissue: nasal swab | DNA/RNA | Respiratory microbiota* |
| | | Direct observation | Anthropometry | Growth |
| | | | Blood pressure | Hypertension |
| | | | Ophthalmology | Refraction and acuity |
| | | | Social development | Parent–child play, for later behavioural coding |
| | | | Executive function | Following Instructions task |
| | | | Developmental level | Mullen Scales of Early Learning |
| | | Eye tracking | Social development | Free scanning: neutral faces |
| | | | | Free scanning: pop-out task, looking to faces and distractors |
| | | | | Free scanning: social preferential looking to social and non-social images |
| | | | | Free scanning: dancing ladies social and non-social videos |
| | | | Attention | Switching and disengagement: gap-overlap task, fixation to central and peripheral cues |
| | | | | Sustained attention: follow the bird task, following moving target |
| | | | | Free scanning: odd-one-out visual search task (complex objects version) |
| | | | Processing speed | |
| | | Questionnaire | Temperament | Strengths and Difficulties Questionnaire (both teacher and parent report versions) |
| | | | Language | Children's Communication Checklist |
| | | | Social development | Social Communication Questionnaire: Current |
| | | | Executive function | DuPaul ADHD Rating Scale |
| | | | | BRIEF-P |
| | | | Visual perception | Cerebral Visual Impairment Inventory |
| | | | Parent well-being | WHO—Quality Of Life |
| | | | Feedback | Feedback form monitoring satisfaction with research project |
| | | | Developmental level | Vineland Adaptive Behaviour Scales: domain-level parent rating form |
| | | Parent interview | Demographics | Family circumstances update form including socioeconomic status (home postcode) |

*Subset of participants.
ADHD, attention deficit hyperactivity disorder; BRIEF-P, Behaviour Rating Inventory for Executive Function, Preschool; HPAA, hypothalamic–pituitary–adrenal axis.

replication studies and investigate risk and resilience in brain development conferred by the genome.[23–25] We expect to address some of these issues with the planned sample of 400 and to make material contribution to wider data-sharing initiatives subject to the study's data access and collaboration policy.

### Identifying participants

Cases: infants born to women who present to the SCRH with threatened preterm labour and for whom delivery is planned or expected at less than 33 weeks GA.

Controls: infants born to women who attend the SCRH and deliver at >37 weeks of GA.

The protocol reported here was partially developed through a separate, pilot 'phase I' cohort of 150 cases and 40 controls. This phase I pilot included neonatal MRI and infant eye tracking, and a subset of this group is now participating in the 5-year assessment as described here (time point 7, table 1).

### Screening for eligibility

The research nurse/clinical research fellow identifies potential participants using maternity TrakCare, which is a system used by maternity services throughout NHS Lothian to record information about pregnancies and maternal care, and the neonatal electronic patient record. The clinical team provides an introductory leaflet about TEBC to eligible parents and then informs the research team of parents who wish to discuss the study in greater detail. Those parents meet with a member of the research team and are provided with the participant information sheet.

Participants from phase I studies being recalled for time point 7 (at 5 years) are contacted by the research team using contact details provided previously. Study information (introductory letter, patient information sheet, reply slip and prepaid envelope) is sent by post and followed up with a telephone call to answer any questions and to review willingness to participate.

### Consenting participants

Informed written consent is sought in two stages: first, consent for perinatal and neonatal sampling and assessment at initial enrolment to the study; second, consent for assessments postdischarge to 5 years is taken at time point 3 (see table 1 below).

For phase I participants being recalled, consent is taken at the recall appointment, following circulation and discussion of the content by post and phone, as described previously.

Informed consent may only be taken by a member of the research team with training in International Council for Harmonisation—Good Clinical Practice and procedures for research involving children and young people.

### Coenrolment

The SCRH is an academic perinatal medicine centre that hosts observational research studies, and it is a recruiting centre for randomised controlled trials of therapies designed to improve the outcome of preterm infants and their mothers. Parents/carers of TEBC participants are encouraged to consider entry into such studies if eligible. Coenrolment is informed by 'Guidelines for Co-enrolment', produced by the Academic and Clinical Central Office for Research and Development (ACCORD), which is a partnership between the University of Edinburgh and NHS Lothian Health Board. Coenrolment will be recorded.

### Cohort retention

Participants and their families are kept up to date with research progress through newsletters, Twitter, Facebook and a website (www.tebc.ed.ac.uk). Birthday cards are sent to participants and we hold an annual event for research updates and public outreach.

### Withdrawal of study participants

The decision to withdraw from the study is either at parental/carer request or at the request of the attending consultant physician or the principal investigator for clinical reasons.

## Outcomes and data analysis

Table 1 summarises the assessment schedule, data collection methods, sample type/domain, and the test or task. Data from cases and controls are collected using the same data collection instruments.

### Maternal and infant clinical and demographic information

Data are abstracted from the mothers' and infants' electronic medical records onto a standardised data collection sheet. A structured maternal interview is used to collect additional information that may not be recorded in routinely collected data, for example, detailed family history about neurodevelopmental and mental health problems, and over-the-counter prescription and recreational drugs taken during pregnancy. For deaths, the cause and postmortem findings will be recorded.

### Placentas

After delivery, placentae from all preterm infants are formalin fixed and stored at 4°C before sampling. The placentae are sampled according to a standardised protocol; distal and proximal sections of cord (the proximal section being taken at 1.5 cm from above the fetal surface), a roll of extraplacental membranes starting at the point of rupture and four full thickness sections from each quadrant. All are stained with H&E and reported using a standardised, structured approach that describes any pathological features present, including but not limited to, fetal thrombotic vasculopathy, villitis, chorioamnionitis, funisitis and features of uteroplacental ischaemia.[12 26]

### Immunoregulatory and trophic proteins

Analysis of a panel of immunoregulatory and trophic proteins (interleukin (IL)-1b, IL-2, IL-4, IL-5, IL-6, IL-8, IL-12, IL-17, tumour necrosis factor (TNF)-α, macrophage

inflammatory protein (MIP)-1b, brain-derived neurotrophic factor (BDNF), granulocyte-macrophage colony-stimulating factor (GM-CSF), IL-10, IL-18, interferon (IFN)-g, TNF-β, monocyte chemoattractant protein (MCP)-1, MIP-1a, C3, C5a, C9, matrix metalloproteinase (MMP)-9, regulated on activation, normal T cell expressed and secreted (RANTES) and C reactive protein (CRP)) is undertaken on umbilical cord and neonatal blood samples. These proteins are selected to offer information with respect to the proinflammatory and anti-inflammatory innate responses, as well as the adaptive immune response. Blood is collected using Schleicher and Schuell 903 filter paper (6.0×3.2 mm spots per subject) and analysed using a multiplex immunoassay (Meso Scale Discovery) at Statens Serum Institute, Copenhagen. We use the approach described by Skogstrand et al[27] to analyse differences in concentration between cases and controls.

## Structural and dMRI

A Siemens MAGNETOM Prisma 3T MRI clinical scanner (Siemens Healthcare, Erlangen, Germany) and 16-channel phased-array paediatric head receive coil are used to acquire three-dimensional (3D) T1-weighted (T1w) magnetization-prepared rapid acquisition with gradient echo (MPRAGE) structural volume scan (acquired voxel size=1 mm isotropic) with inversion time (TI) 1100 ms, echo time (TE) 4.69 ms and repetition time (TR) 1970 ms; a 3D T2-weighted (T2w) sampling perfection with application-optimized contrasts by using flip angle evolution (SPACE) structural scan (voxel size=1 mm isotropic) with TE 409 ms and TR 3200 ms; and a multishell axial dMRI scan (16×b=0 s/mm², 3×b=200 s/mm², 6×b=500 s/mm², 64×b=750 s/mm², 64×b=2500 s/mm²) with optimal angular coverage[28] (online supplementary materials 1–3). If the infant stays settled, axial 3D susceptibility-weighted imaging (TR=28 ms, TE=20 ms, 0.75×0.75×3.0 mm acquired resolution) and axial 2D fluid-attenuated inversion-recovery BLADE imaging (TR=10 000 ms, TE=130 ms, TI=2606 ms, 0.94×0.94×3.0 mm acquired resolution) are acquired. In a subgroup of participants, magnetisation transfer saturation imaging is acquired for evaluation of tissue myelin content, consisting of three sagittal 3D multiecho spoiled gradient echo scans (TE=(1.54, 4.55 and 8.56 ms), 2 mm isotropic acquired resolution), magnetisation-transfer, proton density-weighted (TR=75 ms, flip angle =5°) and T1w (TR=15 ms, flip angle =14°) acquisitions (see online supplementary material 4). Tissue heating and acoustic noise exposure are limited throughout the examination through the use of active noise cancellation and by setting the gradient slew rate and other pulse sequence parameters appropriately. Participants are scanned in normal mode with respect to both tissue heating and peripheral nerve stimulation.

Conventional images are reported by a paediatric radiologist using a structured system.[29 30] We use image data to generate novel processing techniques optimised for neonatal data,[11 19–21 31] and we will use these and other publicly available pipelines for processing neonatal data[13 32 33] to derive image features for analyses with collateral data relating to exposures and outcomes. These include, but are not limited to, tract-based, morphometric and structural connectivity analyses.[10–12 34–38]

## DNA storage

DNA is extracted from saliva, stored and catalogued at the Edinburgh Clinical Research Facility, ready for downstream analyses.

## DNA methylation

Saliva is sampled using the DNA OG-575 kit (DNAGenotek, Ottawa, Ontario, Canada). DNA extraction is performed using published methods,[36] and DNA methylation (DNAm) analyses are carried out at the Genetics Core of the Edinburgh Clinical Research Facility (Edinburgh, UK) using Illumina Infinium MethylationEPIC (San Diego, California, USA), with interrogation of the arrays against ~850 k methylation sites. We will investigate perinatal influences on DNAm using principal component analysis, mediation and correlation analyses.

## Hypothalamic–pituitary–adrenal axis (HPAA)

Salivary cortisol is used as a marker of HPAA activity. Saliva is collected in Sarstedt tubes at specified times at 9 months and 5 years. Timed saliva samples are also collected during the 9-month appointment before and after a behavioural paradigm (still face), which is known to elicit a biological stress response (one sample pretest and two samples post-test to capture reaction and recovery). Samples are stored at −20°C and analysed in batches at each time point. Anthropometric data are recorded at 9 months and 2 and 5 years, and blood pressure is measured at 5 years

## Eye tracking

We record eye movements in response to visual stimuli at 9 months, 2 years and 5 years using a Tobii x60 eye tracker and bespoke analysis software (MATLAB). Images are presented on a display monitor with a resolution of 1440×900 pixels. The Tobii×60 system tracks both eyes to a rated accuracy of 0.3 degrees at a rate of 60 Hz. We analyse looking patterns, including time to first fixate and looking time at areas of interest, in tasks designed to enable inference about social development, attention and processing speed.[35 39]

## Standardised assessments

Standardised assessments of neurodevelopment by direct observation at appropriate time points are Bayley-III scales, Mullen Scales of Early Learning (MSEL) and parental IQ (National Adult Reading Test). We selected the MSEL for assessing cognitive ability at 5 years because it has separate verbal and non-verbal standardised scores, so it is useful for assessing cognitive abilities in children with social communication and language difficulties; internal consistency reliability and test/retest reliability for the five component scales is high; and the early learning composite (and its components) correlate with other psychometric tests used in this age group. We will use validated questionnaires to

assess: infant/parent temperament; parent/family characteristics (postnatal depression, stress, quality of life and socioeconomic status); infant/child sleep habits; language development; social development; executive functions; cerebral visual impairment; medical diagnoses; and behavioural outcomes (parent and teacher ratings). We also record parent–child interaction for subsequent analysis via video coding of complex behaviours in a naturalistic context.

### Susceptibility to viral infection

We collect unstimulated nasal secretion samples (nasosorption samples) using methods described by Thwaites *et al.*[40] This collection is brief, minimally invasive and a minimally distressing process. Nasosorption nasal lining fluid is collected using Nasosorption Fxi synthetic absorption matrix strips inserted into the anterior part of the inferior turbinate of the nasal cavity. After 30s of absorption, the strip is removed, capped, maintained at 4°C for up to 4 hours and then frozen at −80°C. From these nasal fluid samples we will assess the levels of antimicrobial peptides, including cathelicidin, and inflammatory cytokines by ELISA or luminex assay. Collection of these at birth (term equivalent age), 9 months and 2 years will enable us to characterise birth levels, levels at time points significant for respiratory syncytial virus (RSV) infection/disease and at a later time point.

### Respiratory and gut microbiota

We collect faecal and nasopharyngeal swabs (paediatric Copan e-swab with flocked nylon fibre tip) as has been described in the WHO guideline for respiratory sampling of bacterial pathogens.[41] Faecal material and e-swabs (in RNA protect) are frozen at −80°C until further analyses. DNA and RNA will be extracted[42] and metagenomics analyses will be executed by 16S-based sequencing according to previously described methods.[43] We will study temporal relationships between preterm birth and early life characteristics, consecutive microbiota development, inflammation and methylation findings, and respiratory and neurocognitive developmental outcomes.

### Computational motor assessment

Lightweight, wearable, wireless motion sensors are deployed to record the movement of a subset of infants at 9 months during the still-face paradigm and parent–child interaction. Data are anonymised before being securely transferred to the University of Strathclyde for analysis. These data will be analysed to test for differences in motor function between at-risk and low-risk infants, and will employ machine learning algorithms to detect patterns predictive of developmental outcome at 2 and 5 years, and their potential for clinical stratification across the neurodevelopmental disorders and psychometric profiles (IQ, adaptive function and language). Further, motor data at 9 months can be correlated against neuroanatomical features measured by MRI scan at birth and developmental scales at 9 months.

### Patient and public involvement

We seek feedback from parents/carers to monitor satisfaction with research participation at 9 months and 2 and 5 years, and we have a public facing website that describes results from the study.

## ETHICS AND DISSEMINATION

Ethical approval has been obtained from the National Research Ethics Service (NRES), South East Scotland Research Ethics Committee (NRES numbers 11/55/0061 and 13/SS/0143 (phase I) and 16/SS/0154 (phase II)), and NHS Lothian Research and Development (2016/0255). Results are disseminated through open access journals, scientific meetings, social media, newsletters and a study website (www.tebc.ed.ac.uk), and we engage with the University of Edinburgh public relations and media office to ensure maximum publicity and benefit.

### Safety assessment

There are no safety issues associated with collection of: placental tissue, umbilical cord/neonatal blood, saliva, faeces or hair. There are no safety issues in the conduct of planned neuropsychological assessments.

MRI does not involve ionising radiation and there are no known risks from MRI provided standard safety measures for 3T scanning are in place. Infants are fed and wrapped and allowed to sleep naturally in the scanner. Pulse oximetry, electrocardiography and temperature are monitored. Flexible earplugs and neonatal earmuffs (MiniMuffs, Natus) are used for acoustic protection. All scans are supervised by a doctor or nurse trained in neonatal resuscitation. The scan is interrupted if there are any abnormalities in monitoring or if the baby wakes.

It is possible that incidental findings may be found on MRI or from questionnaires, for example, intracranial structural anomalies or postnatal depression, respectively. In these circumstances, the findings are discussed with the participant's parent, and referral to the appropriate NHS service is made.

The study is run by a management group that includes the principal investigator, a minimum of two coinvestigators, the study coordinator and administrative and financial officers. A delegation log details the responsibilities of each member of staff working on the study. A scientific advisory board oversees the conduct and progress of the study. The study is cosponsored by the University of Edinburgh and NHS Lothian ACCORD.

### Publication and data statement

The principles set down by the International Committee of Medical Journal Editors for authorship and non-author contributors are followed for publications and presentations resulting from the study. A data access and collaboration policy sets out the terms and conditions on which deidentified TEBC data, stimuli and tasks are accessible to the research community following reasonable request (www.tebc.ed.ac.uk).

**Acknowledgements** The authors thank participating families and NHS colleagues at the Simpson Centre for Reproductive Health who supported this study. We also thank Mrs Bavanthe Navarathne, a parent representative on the scientific advisory board, and other past and present members of the scientific advisory board (Frances Cowan, Chiara Nosarti, David Porteous, Hugh Rabagliati, Joanna Wardlaw and Heather Whalley). We are grateful to the following collaborators, colleagues and students who supported the study: Gayle Barclay, Justyna Binkowska, Gillian Black, Manuel Blesa, Nis Borbye-Lorenzen, Geoff Carlson, Yu Wei Chua, Simon Cox, Hilary Cruikshank, Bethan Dean, Jonathan Delafield-Butt, Fiona Denison, Margaret Evans, Paola Galdi, Peter Ghazal, Lorna Ginnell, Charlotte Jardine, Gillian Lamb, Victoria Ledsham, Riccardo Marioni, Andrew McIntosh, Barbara Nugent, Lee Murphy, Sinéad O'Carroll, Alan Quigley, Alan Mulvihill, Magda Rudnicka, Scott Semple, Kristin Skögstrand, Sarah Stock, David Stoye, Gemma Sullivan, Kadi Vaher, colleagues at the Genetics Core of the Edinburgh Clinical Research Facility and radiographers at the Edinburgh Imaging Facility Royal Infirmary of Edinburgh.

**Contributors** JPB designed the study with input from all the authors. JPB, JH, MJT, RMR, SC, JS, DB, DJD, AJD, MEB and SF-W contributed to the establishment and refinement of study procedures and critically revised the manuscript. All authors approved the final version of the manuscript.

**Funding** The Theirworld Edinburgh Birth Cohort study is funded by the charity Theirworld (www.theirworld.org) and is carried out in the University of Edinburgh MRC Centre for Reproductive Health (MRC G1002033). Susceptibility to viral infection studies is supported by grants from Action Medical Research (GN2703) and Chief Scientist Office (TCS/18/02). Respiratory microbiota studies are supported by grants from the Chief Scientist Office (SCAF/16/03), and DNA methylation and gut microbiota studies are supported by the Wellcome Trust (203769/Z/16/A and 220043/Z/19/Z). The MRI facility is funded by Wellcome Trust (104916/Z/14/Z), Dunhill Trust (R380R/1114), Edinburgh and Lothians Health Foundation (2012/17), Muir Maxwell Research Fund and Edinburgh Imaging, University of Edinburgh. MJT was supported by NHS Lothian Research and Development Office, and RMR and AJD received support from the British Heart Foundation (RE/18/5/34216).

**Competing interests** None declared.

**Patient consent for publication** Not required.

**Provenance and peer review** Not commissioned; externally peer reviewed.

**Open access** This is an open access article distributed in accordance with the Creative Commons Attribution 4.0 Unported (CC BY 4.0) license, which permits others to copy, redistribute, remix, transform and build upon this work for any purpose, provided the original work is properly cited, a link to the licence is given, and indication of whether changes were made. See: https://creativecommons.org/licenses/by/4.0/.

**ORCID iD**
James P Boardman http://orcid.org/0000-0003-3904-8960

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
