## [Reviewer comments · BMJ Open]

ARTICLE DETAILS

TITLE (PROVISIONAL)	Impact of preterm birth on brain development and long-term outcome: protocol for a cohort study in Scotland
AUTHORS	Boardman, James; Hall, Jill; Thrippleton, Michael; Reynolds, Rebecca; Bogaert, Debby; Davidson, Donald; Schwarze, Jurgen; Drake, Amanda; Chandran, Siddharthan; Bastin, Mark; Fletcher-Watson, Sue

VERSION 1 - REVIEW

REVIEWER	Neil Marlow UCL EGA Institute for Women's Health, London UK
REVIEW RETURNED	02-Dec-2019

GENERAL COMMENTS	This is a protocol for a cohort study which has been active since 2016. It is explorative and therefore it is difficult to estimate the detailed objective as there are many given the comprehensive investigations that are indicated. The stats section is very light saying that there is no established methodology and then basing it on sensitivity analyses for TBSS and structural /diffusion outcomes referring to 9 references and a successful image acquisition of 85%. This could be clearer and give more robust exemplars of indicative power as they surely have done this to come to this conclusion. They will have presumably data from other studies they have done to direct this. It is unclear whether this refers to simply booked deliveries at the Simpson or whether they will use the opportunity of their referral patterns to enrich the population of extremely low gestational age infants. The only other issue is the use of a relatively weak developmental test at 5 years - they should justify not using a more accepted general cognitive score. Otherwise, I find the protocol potentially very exciting and I am sure it will deliver excellent and valuable information into this area.
--

REVIEWER	David Edwards King's College London United Kingdom
REVIEW RETURNED	20-Dec-2019

GENERAL COMMENTS	This is a valuable cohort study established by a major institution with excellent resources, skills and support. It should add significantly to our understanding of the effects of preterm birth on the developing brain.
--

VERSION 1 – AUTHOR RESPONSE

Reviewer(s)' Comments to Author:

Reviewer: 1

Reviewer Name: Neil Marlow

Institution and Country: UCL EGA Institute for Women's Health, London UK

Please state any competing interests or state 'None declared': None

Please leave your comments for the authors below

This is a protocol for a cohort study which has been active since 2016. It is explorative and therefore it is difficult to estimate the detailed objective as there are many given the comprehensive investigations that are indicated.

The stats section is very light saying that there is no established methodology and then basing it on sensitivity analyses for TBSS and structural /diffusion outcomes referring to 9 references and a successful image acquisition of 85%. This could be clearer and give more robust exemplars of indicative power as they surely have done this to come to this conclusion. They will have presumably data from other studies they have done to direct this.

Response 2. We have listed 4 key aims of the cohort study at the end of the introduction, and summarised the corresponding expected outcomes in a summary paragraph at the end of the manuscript. We have clarified the section on sample size with exemplars of indicative power and sensitivity, as requested.

It is unclear whether this refers to simply booked deliveries at the Simpson or whether they will use the opportunity of their referral patterns to enrich the population of extremely low gestational age infants.

Response 3. One of the inclusion criteria is that infants must be born at the Simpson Centre for Reproductive Health because it is not possible to obtain birth samples if delivery takes place outside the study centre. We have clarified processes by adding the following to the Study Participant section:

“Cases are included if a mother booked her pregnancy and delivered at SCRH (the study centre), or if a mother booked her pregnancy at a hospital outside the study centre but was transferred to it with her baby in utero due to planned or expected birth at <33 weeks. Preterm infants who are transferred to SCRH ex utero for intensive care are not included.”

The only other issue is the use of a relatively weak developmental test at 5 years - they should justify not using a more accepted general cognitive score.

Response 4. There are several standardised tools available for assessing cognitive scores at 5 years, all with different strengths and weaknesses. We selected the Mullen Scales of Early Learning for the following reasons:

- It has separate verbal and nonverbal standardized scores (the expressive language scale is the only domain that requires verbal expression), and it includes tasks with low social demands. As such researchers have supported its use for assessing cognitive abilities in children with ASD and

those with social difficulties (Akshoomoff N Child Neuropsych. 2006; Filipek et al J. Autism & Dev. Disord. 1999)

- The median internal consistency reliability for all 5 scales of MSEL is high (0.75-0.83), and test/retest reliability is around 0.8.
- It is recommended to be used as part of a comprehensive battery which includes measures of adaptive and social functioning, which are a specific focus in our schedule of assessment.
- Validity has been demonstrated through correlations between the Early Learning Composite and the Mental Development Index on the Bayley Scales of Infant Development (.70). The Mullen Receptive Language scale correlates .85 with the Preschool Language Assessment Auditory Comprehension and .72 with verbal ability, and the Mullen Expressive Language scale correlates .80 with verbal ability and .72 with auditory comprehension. The Mullen Fine Motor scale is strongly correlated with the Peabody Fine Motor Scale: correlations range from .65 to .82, depending on age (Encyclopedia of Clinical Neuropsychology, 2011 ed.)
- The MSEL meets the federal mandate for preschool assessment, and is frequently used to determine eligibility for early intervention services, so it may have wider utility in the target population.

We have added the following sentence to the section 'standardised assessments':

"We selected the MSEL for assessing cognitive because: it has separate verbal and nonverbal standardised scores so is useful for assessing cognitive abilities in children with social communication and language difficulties; internal consistency reliability and test/retest reliability for the 5 component scales is high; and the early learning composite (and its components) correlate with other psychometric tests used in this age group."

Otherwise, I find the protocol potentially very exciting and I am sure it will deliver excellent and valuable information into this area.

Response 5. Thank you for noting the potential for excellent and valuable information.

Reviewer: 2

Reviewer Name: David Edwards

Institution and Country:

King's College London

United Kingdom

Please state any competing interests or state 'None declared': None

Please leave your comments for the authors below

This is a valuable cohort study established by a major institution with excellent resources, skills and support. It should add significantly to our understanding of the effects of preterm birth on the developing brain.

Response 6. Thank you for the positive comments about the value of the study.

VERSION 2 – REVIEW

REVIEWER	Neil Marlow UCL London UK
REVIEW RETURNED	23-Jan-2020

GENERAL COMMENTS	Thank you for addressing my comments. It is an ambitious study but with a likely high return. I have no further comments.
---